# Health Care Professionals’ Perspectives on Life-Course Immunization: A Qualitative Survey from a European Conference

**DOI:** 10.3390/vaccines8020185

**Published:** 2020-04-14

**Authors:** Roy K. Philip, Alberta Di Pasquale

**Affiliations:** 1Division of Neonatology, Department of Pediatrics, Graduate Entry Medical School (GEMS), University of Limerick and University Maternity Hospital Limerick (UMHL), V94 C566 Limerick, Ireland; 2Medical Department, W23, 20 Avenue Fleming, 1300 Wavre, Belgium; alberta.di-pasquale@gsk.com

**Keywords:** life-course immunization (LCI), healthy aging, vaccine confidence, vaccination behavior, audience response system (ARS) survey

## Abstract

Today, fewer children die each year from vaccine-preventable diseases than older adults. Health systems need new immunization strategies to tackle the burden of vaccine-preventable disease in an aging society. A life-course immunization (LCI) approach—which entails vaccination throughout an individual’s lifespan—enables adults to age with reduced risk to disease, thereby enabling healthy, active and productive aging. We conducted an audience response system (ARS)-based survey to investigate HCP perspectives on LCI in an opportunistic sample of 222 health care professionals (HCPs) from around the world who attended a European infectious diseases conference. Survey results show that LCI is a priority for HCPs (77.4%–88.6%), with most of them stating the need to frame it as a part of a healthy lifestyle (91.0%–100.0%). Insufficient LCI recommendations by vaccine providers (12.9%–33.3%) and governments (15.2%–41.9%) and insufficient targeted budget allocation (6.1%–21.7%) were indicated as the main barriers to implement LCI, ahead of vaccine hesitancy (9.7%–15.2%). HCPs were willing to make LCI a gateway to healthy aging but need support to work together with other stakeholders involved in the vaccination journey. This could be a step towards equitable health care for all of society.

## 1. Introduction

Worldwide, average life expectancy has rapidly increased from 66.5 years in 2000 to 72.0 years in 2016, with an increasing trend predicted for the next decades [1]. By 2030, almost 1 billion people will be over 65 years of age and, for the first time in history, this age group will outnumber children below the age of five years [2]. There is no precedent in history for a society with this demographic structure and health care systems will need new policies to provide care for a society with an aging profile [2]. The World Health Organization (WHO) has recommended health-promotion and disease-prevention strategies to maintain the health and independence of this aging population [3]. Strategies such as national immunization programs have contributed greatly to the increase in life expectancy by reducing the burden of infectious diseases over the last century [4]. Despite these efforts, immunization rates in adolescents and adults remained below recommended targets in many countries [5]. Specifically, the uptake and optimization of maternal immunization—vaccination which protects pregnant women and newborns against vaccine-preventable diseases—deserves special attention [6]. Maternal immunization is the only immunization strategy that directly benefits two generations through a single preventive intervention. Still, the worldwide implementation of maternal immunization has remained suboptimal [7]. Furthermore, the WHO also recommends routine immunization for healthy adolescents (e.g., human papillomavirus vaccine for adolescents) and adults (e.g., seasonal influenza vaccine), timely receipt of the booster dose (e.g., tetanus toxoid, reduced diphtheria toxoid and acellular pertussis [Tdap] and meningitis) and catch-up vaccinations if routine immunization during childhood was missed (e.g., the measles, mumps, rubella or varicella vaccine) [8]. Despite these recommendations, global vaccine uptake remains below recommended levels [9,10].

Considering this situation, experts and institutions recommend extending the vaccine prevention strategy to a “life-course” approach that places emphasis on the need for vaccination against vaccine-preventable diseases through all stages of an individual’s lifespan [11]. This vaccination strategy is called life-course immunization (LCI) [12,13]. The LCI approach enables individuals to age better by reducing the burden of vaccine-preventable infectious diseases [12]. Despite this significant potential and the endorsement by the WHO, LCI has not yet become a priority on national health agendas [14]. A limitation of this approach could be that since the term is loosely defined, the benefits of adopting LCI may not be easily perceived or may depend on the specifics of the vaccination program [15]. Barriers to LCI include the lack of established vaccination delivery strategies in all age groups and the negatively perceived safety and effectiveness of vaccination [16]. Low vaccine coverage and a lack of vaccine knowledge (beyond childhood vaccination) among the general public and health care professionals (HCPs) create additional barriers [16]. Published literature suggests that HCP attitudes and recommendations play an important role in vaccination uptake among individuals of all ages [12]. Therefore, it is crucial to understand their attitudes towards LCI. We conducted an audience response system (ARS)-based survey to investigate HCP perspectives on LCI.

## 2. Materials and Methods

This work was presented in part at a satellite symposium on LCI held at the 35th European Society of Paediatric Infectious Disease (ESPID) conference in Madrid, Spain in 23–27 May 2017. We conducted a survey of an opportunistic sample of 222 health care professionals (HCPs) from around the world, who were among the attendees of the satellite symposium on LCI.

The LCI symposium participants were not asked about their locations of origin. Hence, demographic information on the participants could only be estimated from the overall conference participants. Out of 3039 overall conference participants, the majority came from Europe (68%) and East Asia and Pacific (10%) regions. The remaining participants were from Central and South America (7.0%), Middle East (5.0%), North America (5.0%), Africa and Atlantic (2.0%) and Central Asia (2.0%). Spain (18.0%) and United Kingdom (11.0%) were the most represented countries among the participants of the conference. Other countries represented among the participants were France, Belgium, Greece, Portugal, Romania, Lebanon, Germany, Brazil, Italy, Pakistan, Russia, Switzerland, The Netherlands, Austria, Bangladesh, India and Turkey.

Professional interests of the conference participants were represented under five categories—namely, pediatrics (43.0%), pediatric infectious diseases (38.0%), infectious disease (8.0%), public health and preventive medicine (7.0%) and microbiology (4.0%). Participants identified their professional roles as clinical practitioner (35.0%), resident/research fellow (14.0%), clinical researcher (14.0%), industry/corporate professional (13.0%), other (10.0%) and student (10.0%). The remaining participants identified basic science researchers (3.0%) and nurse/health care practitioners (1.0%) as their professional role.

### 2.1. Participants

An open invitation to attend the satellite symposium at the ESPID 2017 conference was circulated through the conference website and at the conference venue. Out of the 3039 individuals who attended the ESPID 2017 conference, 222 HCPs participated in the symposium. There were no exclusion criteria for the HCPs participating in the survey and participation was voluntary. However, 144 industry representatives who participated in the survey were excluded from the post-event analysis. No incentives were offered for attendance or participation in the survey. Consent to participate in the anonymized survey and to use the collected data for analyses was obtained at the outset of the electronic survey. Participants responded to questions that appeared on the central display, after the lectures of experts in the field of LCI and prior to the open Question & Answer session (Q&A).

No personal details were asked for or recorded other than the participant’s profession. The question on participant profession was asked as a multiple-choice question with five choices (pediatrician/pediatrician subspecialist, infectious disease clinician, public health professional, industry representative or other). Possibility to provide an answer to explain the “other” as a profession was not provided in the survey. While cognizant of the potential for an overrepresentation of pro-vaccination HCPs among the survey subjects, we found it useful to collect their structured feedback on LCI as it could add value to the challenges and opportunities in the implementation of LCI. Even though we did not collect information on survey subjects’ nationality or country of clinical practice, although multiple nationalities from both developed and developing regions of the world attended the European conference. We assumed proportional representation among those who attended the symposium at which the survey was conducted.

### 2.2. Survey Administration

The survey was conducted on 23 May 2017 just after the symposium presentations and prior to the open Q&A. The topic of LCI was relatively new for the ESPID audience composed mainly of pediatricians treating only children, and therefore it was of interest to gauge their view on the life-course approach. This survey was performed after the presentations to make sure participants had understood what LCI was about even though this could have introduced a positive bias. Each participant was given an electronic voting pad to interactively answer survey questions. The survey instrument consisted of multiple-choice questions administered in English and aimed to collect data on HCP perspectives on LCI. The survey assessed whether LCI was a priority for the participants, their views on LCI as part of a healthy lifestyle, their willingness to spread awareness about LCI and to engage with other health care specialists to discuss LCI. The survey also assessed participants’ perspective on possible barriers to implement LCI and the role of industry in spreading awareness about LCI. Survey questions were developed by the panel of speakers from various health care backgrounds including an epidemiologist, a public health expert, a pediatrician and an anthropologist. The intent was not to achieve a generalizable outcome but to gather insights and the immediate reaction of the international audience of HCPs.

The ARS used for the real-time survey was the OMBEA response system (OMBEA Ltd., Covent Garden, London, UK) [17].

### 2.3. Survey Questions, Data Collection and Statistical Analysis

The electronic questionnaire aimed to highlight HCP insights related to LCI based on three topics of interest, namely LCI perceptions and role, challenges for LCI, and the role of industry. First, LCI perceptions and role were evaluated by asking the following questions (1) Is life-course immunization a priority for you? (2) Are you ready to discuss life-course immunization with your patients and recommend it for the entire family? (3) Are you willing to engage with other HCPs (specialists) to spread knowledge about the importance of vaccination for all ages? and (4) Do you think it is useful to frame life-course immunization as part of a healthy lifestyle (together with diet, physical exercise, smoking cessation)? Second, challenges for LCI were evaluated by asking the following questions (1) In your profession, what do you encounter as the main reason for vaccine hesitancy? and (2) What do you think is the main barrier to achieving high coverage at all ages? Third, the role of industry was evaluated through the following questions (1) Do you think that the industry should be an active partner? and (2) In addition to supplying vaccines, what should industry prioritize to stimulate life-course immunization?

Data collected from the real-time electronic questionnaire were analyzed and presented live as overall results, while post-event analyses were conducted by profession. Descriptive statistics were presented using the numbers and percentages of responses for each question. The overall results of the votes to each question were visible to the participants and to the faculty of the symposium which supported the live discussion on LCI. Representatives from the industry who participated in the survey were excluded from the analysis since the objective of the research was to understand the perspective of HCPs. Not all participants answered all questions and therefore the number of responses by each profession was different.

## 3. Results

A total of 222 HCPs from various medical backgrounds including pediatricians (n = 104), infectious disease specialists (n = 40), public health professionals (n = 39) and others (n = 39) who attended the symposium were surveyed.

### 3.1. LCI Perceptions and Role

Among all surveyed participants, 77.4%–88.6% indicated that LCI was a priority. A few participants indicated that they did not know about LCI, while 10.0%–22.6% indicated that childhood vaccination was still the priority for them (Figure 1).

Most participants indicated that they were ready to discuss LCI with their patients and extend the recommendation for LCI to the patient’s family (59.4%–75.7%). Among all professions, between 12.5% and 23.7% of participants indicated that they needed additional support to have discussions about LCI with their patients. Among the three HCP specialties, pediatricians represented the highest percentage needing some support (23.7%; Figure 2).

Among all participants, 51.7%–63.9% stated that they were willing to engage with HCPs from other specialties to spread knowledge about the importance of vaccination for all ages. Participants also indicated that they had too little interdisciplinary contact (13.9%–27.6%) or needed support to engage with HCPs from other specialties (13.8%–25.0%; Figure 3).

A clear majority of participants “strongly agreed” or “agreed” on the need to position LCI as part of a healthy lifestyle together with diet, physical exercise and smoking cessation (91.0%–100.0%; Figure 4).

### 3.2. Challenges for LCI

Participants indicated that they encounter vaccine hesitancy in their practice and the main reasons for this were safety concerns about vaccines (23.3%–43.3%), negative influence by social media (18.2%–36.7%) and insufficient disease awareness in parents/individuals who are involved in making decisions about vaccination (13.3%–28.8%; Figure A1).

Vaccine hesitancy was not the main barrier to achieving optimal vaccination coverage levels as only 9.7%–15.2% of participants indicated that it is an impediment in the context of LCI. Other barriers to achieving optimal vaccination coverage are insufficient recommendations by vaccine providers (12.9%–33.3%), insufficient government recommendations (15.2%–41.9%) and insufficient budget allocated to LCI (6.1%–21.7%; Figure A2).

### 3.3. The Role of Industry

Most participants indicated that vaccine manufacturers should play an active role in supporting LCI, with 66.7%–80.8% agreeing that their role should be more active, while 7.7%–24.2% disagreed, and 3.7%–11.5% indicated that they did not know what role the industry should play (Figure A3).

When asked about the initiatives to be prioritized by vaccine manufacturers, participants indicated educational activities (35.3%–57.7%) as the most important, followed by access to scientific information and innovation (26.9%–52.9%), addressing vaccine confidence/hesitancy, (2.9%–15.4%), and others (0.0–8.8%; Figure A4).

## 4. Discussion

We conducted an ARS-based survey to investigate HCP perspectives on LCI. By surveying an opportunistic sample of participants from various health care backgrounds, we were able to gain insights from the frontline vaccinators. We gained firsthand knowledge about the importance of LCI among HCPs, the problems HCPs face in implementing LCI and the role of industry in supporting HCPs in implementing LCI. The key message of our survey is provided in Figure 5.

The novelty of this survey is to collect insights generated by documenting HCP views on LCI. HCPs agreed that, together with healthy diet and exercise, LCI could help achieve health and well-being of individuals at all ages throughout their lifespan [16,18]. The focus of health systems is too often limited to the programmatic issues of implementing health care interventions and the opinions of frontline vaccinators are rarely considered. This paper shares HCP insights on LCI which we see as an important addition to available evidence, since such survey data are seldom published besides the live interaction [12,19,20]. Furthermore, pedagogical tools such as ARS have demonstrated value by facilitating active learning and ‘sense making’ through real-time insights, hence supporting our rationale for administering an ARS-based survey [7,21].

In this survey, 88% of pediatricians indicated that they viewed LCI as a priority. Interestingly, infectious disease and public health specialists also viewed LCI as a priority (77%–88%). This may be due to the fact that older populations are more susceptible to infectious diseases and the participants might see interventions such as LCI as a viable solution for this population [2]. We also found that interdisciplinary contact among HCPs is currently lacking, as participants indicated that they had too little interdisciplinary contact (13.9%–27.6%) or needed support to engage together (13.8%–25.0%). Regardless of specialty (pediatrician, infectious disease specialist or public health professional), participants specified the need to collaborate with other specialties and frame LCI as part of a healthy lifestyle to their patients and extend such discussions to their families. These results encourage the need to disseminate knowledge about the importance of vaccination for all ages through increased interdisciplinary contact among HCPs [22].

Participants in this survey were motivated to spread awareness about LCI among other HCPs, patients and families. However, the reasons why practitioners often fail to deliver on their apparent enthusiasm for the importance of LCI was not captured through this survey but is potentially valuable in understanding the barriers related to LCI. This was part of our previous related work in which we conducted a focus group study asking HCPs to describe their expectations as frontline vaccinators versus the day-to-day reality they faced [11]. Participants from the focus group described challenges impacting their role as vaccinators and proposed key solutions to these challenges. Sixteen groups of HCPs from the US, Germany, UK and India, comprising a total of 75 nurse and physician vaccinators, participated in individual and focus group discussions in 2018 [11]. Individual and focus group responses were analyzed following narrative analysis principles. Uncertainty surrounding current immunization guidelines, cost of vaccines, recurring vaccine stock shortages and misinformation about vaccines were all highlighted as challenges for the frontline HCPs [11]. These results emphasize the need to support vaccinators in their role to ensure that they can continue delivering on the success of vaccination programs and integrate the LCI strategy.

We found that participants viewed the need for vaccine manufacturers to play an active role in educating HCPs and other stakeholders in order to stimulate dialogue around the need for LCI. Overall, this survey shows the need for all stakeholders involved in the process of vaccination to work together to ensure that people may have long and healthy lives through LCI. This is in line with previous publications which recommend a multi-disciplinary approach involving all relevant stakeholders to ensure the successful implementation of LCI [11,12]. As an example, a similar cooperative approach achieved success in Italy, where a multi-disciplinary partnership of medical scientific societies representing public health, primary care and pediatrics successfully collaborated to produce three consecutive editions of the “Lifetime Immunization Schedule” [23]. An implication of this result is that industry could facilitate knowledge-sharing initiatives about LCI through multi-disciplinary partnerships with HCPs and other relevant stakeholders.

In this survey, vaccine hesitancy was not seen as the main barrier in the implementation of LCI. However, vaccine hesitancy is commonly documented in literature as a barrier to immunization [24,25]. As such, vaccine hesitancy deserves attention, especially since the WHO considers it a major threat to global health [10]. The WHO states that vaccine hesitancy is a phenomenon in the developed and developing countries alike as evidenced from the reemergence of infectious diseases which had previously been eradicated or controlled [26]. These results warrant the need for further research to understand the importance of vaccine hesitancy as a major factor affecting global health and to better understand its role as a barrier to LCI implementation.

Strengths of this research include that survey participants came from both the developed and the developing regions of the world and, as such, results from this study could provide insights into transferability to other settings, which is in line with results from a prior study by Wiot et al. [11]. Furthermore, this paper shares HCP insights on LCI which may be an important addition to the presently available evidence on LCI. This is because such survey data are seldom published and occasionally overlooked for programmatic issues of implementing health care interventions.

We acknowledge certain limitations in our survey administration, and therefore the findings may not be generalizable to the wider group of HCPs. This can be partly attributed to the small sample size of the groups surveyed. Attendance to this symposium at a conference on pediatric infectious disease followed by the survey being conducted after the speakers presented their perspective on LCI implies selection bias that would drive answers towards a more positive trend to the topics/questions. Stratified analysis beyond profession or assessment for possible biases would not be possible due the small sample size of each stratum. Such an exploratory in-depth analysis is also not logistically feasible during the limited cross-sectional timeframe to gather data through an ARS-based system amidst an international conference. The majority of conference attendees were from developed European countries. This could lead to various biases, since opinions of HCPs from developed countries could be different than those from developing countries. Another limitation of this study is that LCI is not a defined entity and the overall benefits of the approach may depend on the specifics of the immunization program such as the vaccines being considered, immunization schedules, number of doses, population characteristics and health economic evaluations. Lastly, health economic evaluations are increasingly important to compare immunization programs against each other, and the importance of such research was not evaluated in the survey.

## 5. Conclusions

This survey aimed to capture HCP perspectives on LCI and understand the barriers they faced in LCI implementation and the role of industry in supporting HCPs to implement LCI. Our findings suggest that LCI is viewed as a priority by HCPs involved in the vaccination journey and there is agreement among HCPs to position LCI together with diet and physical exercise to achieve a healthy lifestyle. The results of this survey suggest a willingness of HCPs to work together to make LCI a gateway to ‘healthy aging’ for all people. Simultaneously, the results also shed light on various barriers to vaccination such as lack of provider and government recommendations, insufficient infrastructure and insufficient budget ahead of vaccine hesitancy. We found that education and information campaigns for HCPs and patients could have a tangible impact on the implementation of LCI. Our ARS-based, cross-sectional survey of a global sample of HCPs offers an innovative way to capture insights that could instigate further research to analyze how LCI is perceived by vaccine providers. The results of such studies could further support health care policy makers to develop more comprehensive and effective public health solutions.

## Figures and Tables

**Figure 1 vaccines-08-00185-f001:**
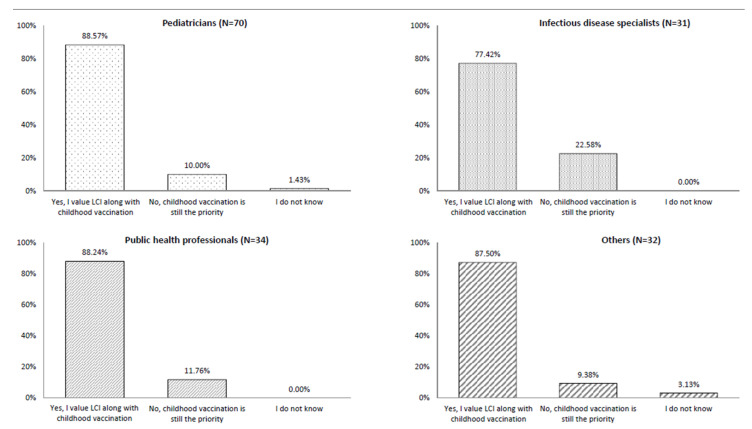
Is life-course immunization a priority for you? Since all participants did not answer all questions, the number of responses by each profession was different from the overall number of participants in each professional category.

**Figure 2 vaccines-08-00185-f002:**
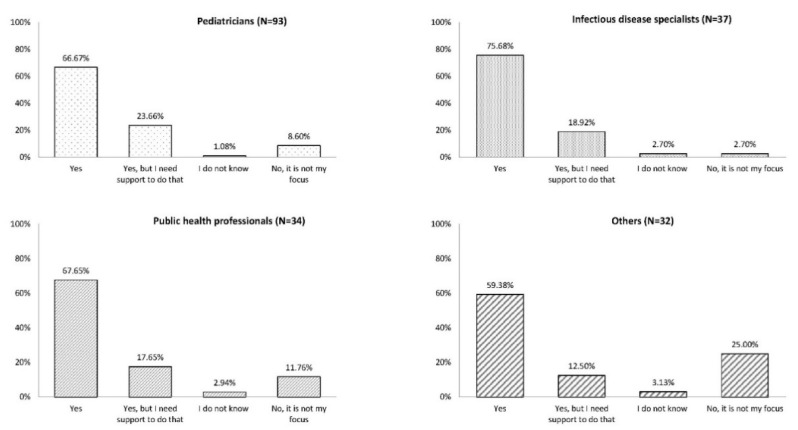
Are you ready to discuss life-course immunization with your patients and recommend it for the entire family? Since all participants did not answer all questions, the number of responses by each profession was different from the overall number of participants in each professional category.

**Figure 3 vaccines-08-00185-f003:**
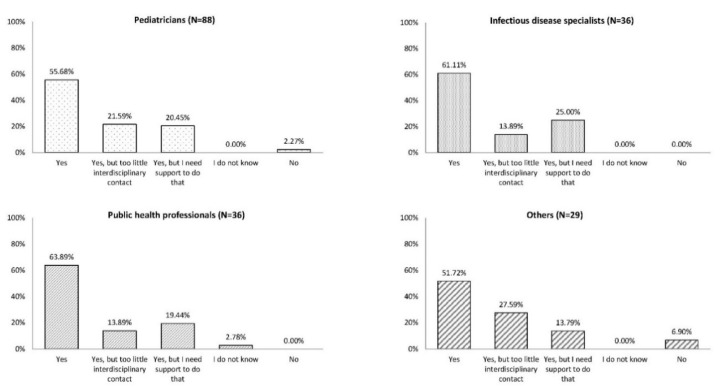
Are you willing to engage with other HCPs (specialists) to spread knowledge about the importance of vaccination for all ages? HCP (specialists) refers to HCPs from other health care specialties in addition to their own profession. HCP, health care professional. Since all participants did not answer all questions, the number of responses by each profession was different from the overall number of participants in each professional category.

**Figure 4 vaccines-08-00185-f004:**
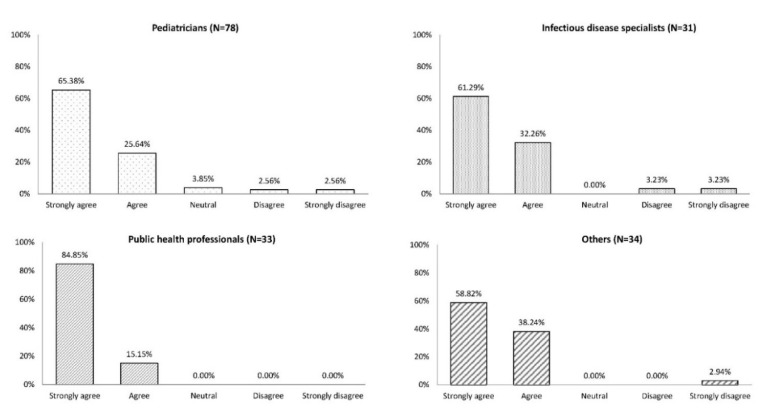
Do you think it is useful to frame life-course immunization as part of a healthy lifestyle (together with diet, physical exercise, smoking cessation)? Since all participants did not answer all questions, the number of responses by each profession was different from the overall number of participants in each professional category.

**Figure 5 vaccines-08-00185-f005:**
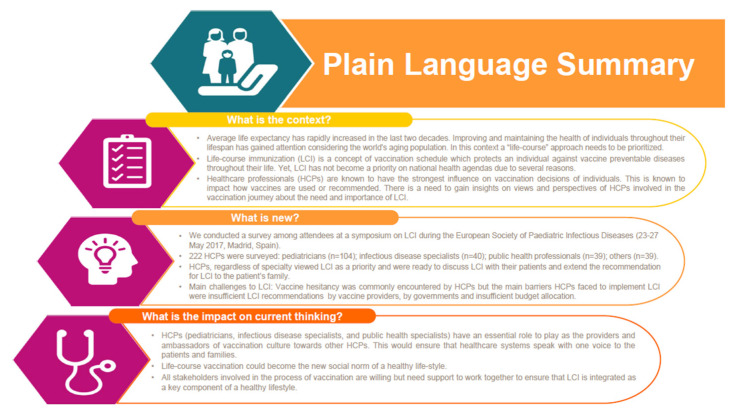
Key message of the ARS-based survey.

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
