# Peer review of "Health Care Professionals’ Perspectives on Life-Course Immunization: A Qualitative Survey from a European Conference"

_vaccines, 2020, doi:10.3390/vaccines8020185_

Round 1

Reviewer 1 Report

The manuscript requires minor editing of grammer and formatting. Appreciating that the authors intended only to canvass attitudes and opinions and that the response rate is low, I offer the following for your consideration:

The authors make several claims in the text which do not appear t be supported by their evidence for example: "...the results on this survey shed light on vaccine misinformation" (line 297), but I can see no evidence of the fact that they do, though one question is asked about vaccine hesitancy.

The statement "...in order to achieve improvements in public health policy" (line 214) regrettably doesn't lead to any concrete suggestions as to how this might be achieved, especially as they relate to the issues faced by 'frontline vaccinators' which the authors refer to.

The work I think raises more questions than it answers but I support the authors contention that more work needs to be done to activate the apparent high willingness of HCPs to respond to this issue.

The work is heavily biased by the timing of the survey ie: participants are primed by the information provided by speakers immediately prior, however the authors do acknowledge this limitation. Informing public policy corrections and improved public health action would require far more rigorous evidence

Author Response

Referee 1 comments
1. The manuscript requires minor editing of grammar and formatting. Appreciating that the authors intended only to canvass attitudes and opinions and that the response rate is low, I offer the following for your consideration:
Author response: Thank you for allowing us to implement these minor changes. We have revisited the entire manuscript for English language and style as requested by the reviewer and made appropriate edits.

2. The authors make several claims in the text which do not appear to be supported by their evidence for example: "...the results on this survey shed light on vaccine misinformation" (line 297), but I can see no evidence of the fact that they do, though one question is asked about vaccine hesitancy.
Author response: Thank you for your feedback. We have revised the text and the conclusions section to specifically mention the points highlighted by the results of the study. Pg. 10, Line 311-312, and Line 316-318.

3. The statement "...in order to achieve improvements in public health policy" (line 214) regrettably doesn't lead to any concrete suggestions as to how this might be achieved, especially as they relate to the issues faced by 'frontline vaccinators' which the authors refer to.
Author response: Thank you for your feedback. We have removed the highlighted text and maintained the focus on issues directly related to frontline vaccinators such as the successful implementation of Life-course immunization (LCI). Pg. 8, Lines 233-236.

4. The work I think raises more questions than it answers but I support the authors contention that more work needs to be done to activate the apparent high willingness of HCPs to respond to this issue.
Author response: Thank you for your feedback. We appreciate the reviewer’s willingness to support our perspective on Healthcare professionals’ (HCP) attitudes towards LCI. This needs to be further highlighted by other researchers in order to overcome the paucity of literature in this subject so that public health systems can take this information into account while developing healthcare interventions. Due to the nature of our survey we could not delve deeper into the subject of stratified analysis as mentioned in the limitations section of this manuscript. Therefore, we accept that many questions arise from this research, but it is our hope that other researchers build upon this research.

5. The work is heavily biased by the timing of the survey ie: participants are primed by the information provided by speakers immediately prior, however the authors do acknowledge this limitation. Informing public policy corrections and improved public health action would require far more rigorous evidence.
Author response: Thank you for your feedback. Indeed, the reviewer’s point about bias has been clearly highlighted in the limitations section of this manuscript. We agree that rigorous evidence would be required for improved public health action. To that end, by focusing on HCP perspectives on LCI we hope that other researchers will add to literature on this and other related subjects, which may lead to these topics informing public health policy decisions.

Reviewer 2 Report

General

Authors explore in their manuscript ‘Health care professionals’ perspectives on life-course immunization:

results of a qualitative survey from a European conference’ the health care professionals' opinion on life-course immunization. Such studies are scarce, however some work should be done before this manuscript can be published.

Title

Please delete from the title at least: <Results of>.

In my opinion the title is lengthy, and should be shortened.

Abstract

Background

Please explain what you mean with <in this demographic>

< … aging.> Please add the aim of the study. (aim Intro is: We conducted an audience response system (ARS)-based survey to investigate HCPs perspectives on LCI.)

Methods

< … to capture their perspectives on LCI.> How did you measure these?

Results

-

Conclusion

-

Key words

Please consider the last keyword; at least it is not in the title, or in the Abstract itself.

Introduction

50           <Average life expectancy has rapidly increased from 66.5 years in 2000 to 72.0 years in 2016>

about which country or region are you talking here? The reader has the right to know this.

Methods

Sample

96           delete (or explain to me why this is necessary info) <followed by France (8.0%), 96 Belgium (7.0%), Greece (6.0%) and Portugal (5.0%). Participants from Romania, Lebanon and Germany comprised of 4.0% of the cumulative conference population while Brazil, Italy, Pakistan, Russia, Switzerland and The Netherlands comprised of 3.0% cumulatively. The remaining 2.0% of the participants represented Austria, Bangladesh, India and Turkey.>

It would be better to mention here how your 222 visitors of the satellite symposium (your real sample) is divided over regions of the world.

113         <representatives from the industry within the audience, …, were excluded from the post-event analysis> Please add how many these were.

135         please change <thought his could> into <thought this could>

Measures

138         please consider adding this heading

Statistical analyses

Please rewrite this section: First, we … . Then, we … . Next, we … . Finally, we … . The readership easier grabs what you did and in which order the Results will be shown.

Results

Figure 1 consider of making: <Yes, I value LCI along with childhood vaccination> (4x)

How is it possible that numbers differ between the text and fig 1 (eg pediatricians N=109 – N=70)

172         Consider changing <percentage (23.7%; Figure 2).> into <percentage needing some support (23.7%; Figure 2).>

Discussion

Please harmonise the research question in the Abstract, in the Introduction section, and the first line of the Discussion section.

Please keep in mind the following structure for writing a Discussion:

Para1                     start with repeating the research question + answer this without any comments or interpretation.

Please rewrite.

Para2,3,#              start a new para, 1 topic per para, and start this para with one of your findings (We found …) – which then defines the content of the para. Relate your finding to earlier published references.

213         unclear which finding you are discussing

222         clear

234         clear

246         unclear which finding you are discussing

259         unclear which finding you are discussing

Strengths and limitations

Mention strengths! Condense a bit limitations.

Implications

(split into: for practice, for future research)

Conclusion

-

Please rewrite this part

Tables, Figures

Please have a look in the Results.

References

Please harmonize the (accessed on Nov 21) and (accessed on 21 September 2017) all over the references.

Be consistent in using abbreviations for journal’s names.

Author Response

Referee 2 comments
Authors explore in their manuscript ‘Health care professionals’ perspectives on life-course immunization: results of a qualitative survey from a European conference’ the health care professionals' opinion on life-course immunization. Such studies are scarce; however some work should be done before this manuscript can be published.
1. Title: Please delete from the title at least: <Results of>. In my opinion the title is lengthy, and should be shortened.
Author response: Thank you for your feedback. We have edited the title as per the reviewer suggestion.
2. Abstract, Background: Please explain what you mean with <in this demographic>.
< … aging.> Please add the aim of the study. (aim Intro is: We conducted an audience response system (ARS)-based survey to investigate HCPs perspectives on LCI.)
Author response: Thank you for your feedback. Text changes implemented on Pg. 1, Line 32 and 34.
3. Abstract, Method: < … to capture their perspectives on LCI.> How did you measure these?
Author response: Thank you for your feedback. This text has been removed from the abstract as the aim of the study clearly demonstrates how we conducted a survey to investigate HCP perspectives on LCI. Pg. 1, Line 34-35.
4. Keywords: Please consider the last keyword; at least it is not in the title, or in the Abstract itself.
Author response: Thank you for your feedback. We have revised the keywords as suggested by the reviewer. Pg. 2, Line 46-47.
5. Introduction: <Average life expectancy has rapidly increased from 66.5 years in 2000 to 72.0 years in 2016>. About which country or region are you talking here? The reader has the right to know this.
Author response: Thank you for your feedback. We have edited the text to specify that the data represents global trends. Pg. 2, Line 50.
6. Methods, Sample: Delete (or explain to me why this is necessary info) <followed by France (8.0%), 96 Belgium (7.0%), Greece (6.0%) and Portugal (5.0%). Participants from Romania, Lebanon and Germany comprised of 4.0% of the cumulative conference population while Brazil, Italy, Pakistan, Russia, Switzerland and The Netherlands comprised of 3.0% cumulatively. The remaining 2.0% of the participants represented Austria, Bangladesh, India and Turkey.> It would be better to mention here how your 222 visitors of the satellite symposium (your real sample) is divided over regions of the world.
Author response: Thank you for your feedback. We have revised the text to explain why information regarding participant area of origin was necessary to be included in the methods section. We have also condensed the text to reduce the amount of information being presented on this subject. Pg. 3, Line 92-94 and 98-100.
7. Methods, Sample: <representatives from the industry within the audience… were excluded from the post-event analysis> Please add how many these were.
Author response: Thank you for your feedback. Text has been edited to reflect the number of industry representatives in the survey. Pg. 3, Line 113.
8. Methods, Sample: Please change <thought his could> into <thought this could>
Author response: Thank you for your feedback. We have implemented the edit as requested by the reviewer. Pg. 4, Line 135.
9. Methods, Measures: Please consider adding this heading - Statistical analyses
Author response: Thank you for your feedback. We have implemented the edit as requested by the reviewer. Pg. 4, Line 148.
10. Methods, Measures: Please rewrite this section: First, we …. Then, we …. Next, we …. Finally, we …. The readership easier grabs what you did and in which order the Results will be shown.
Author response: Thank you for your feedback. We have edited the section with text which clearly specifies for the reader how the results will be presented. Pg. 4, Line 149-161.
11. Results: Figure 1 consider of making: <Yes, I value LCI along with childhood vaccination> (4x).
Author response: Thank you for your feedback. We have implemented the change in text as requested by the reviewer. Pg. 6, Line 218.
12. Results: How is it possible that numbers differ between the text and fig 1 (eg pediatricians N=109 – N=70)
Author response: Thank you for your feedback. Since all participants did not answer all questions, the number of responses by each profession was different from the overall number of participants in each professional category. This explanation has been added as a footnote to all the figures in the manuscript text and appendix. Pg. 6-7 and 11-12.
13. Results: Consider changing <percentage (23.7%; Figure 2).> into <percentage needing some support (23.7%; Figure 2).>
Author response: Thank you for your feedback.We have implemented the change as requested by the reviewer. Pg. 6, Line 226.
14. Discussion: Please harmonies the research question in the Abstract, in the Introduction section, and the first line of the Discussion section.
Author response: Thank you for your feedback. We have implemented the changes as requested by the reviewer. Pg. 1 and 8, Line 34-35 and 229.
15. Discussion: Please keep in mind the following structure for writing a Discussion: Para1 start with repeating the research question + answer this without any comments or interpretation.
Author response: Thank you for your feedback. We have edited the discussion section as requested by the reviewer. Pg. 8, Line 229-233.
16. Discussion: Please rewrite – Para 2, 3, # start a new para, 1 topic per para, and start this para with one of your findings (We found …) – which then defines the content of the para. Relate your finding to earlier published references.
Author response: Thank you for your feedback. We have edited the particular paragraph within the discussion section as requested by the reviewer. Pg. 8, Line 234-237.
17. Discussion: 213 unclear which finding you are discussing.
222 clear
234 clear
246 unclear which finding you are discussing
259 unclear which finding you are discussing.
Author response: Thank you for your feedback. We have edited the section as requested by the reviewer. Pg. 8, Line 234-237, Line 255-258, Line 280-282 and Line 263.
18. Discussion, Strengths and Limitations: Mention strengths! Condense a bit limitations.
Author response: Thank you for your feedback. We have added strengths to the section and condensed the limitations as requested by the reviewer. Pg. 8, Line 289-294.
19. Discussion, Implications: (split into: for practice, for future research)
Author response: Thank you for your feedback. We have added implication to each one of our results highlighted in the discussions section. Pg. 8, Line 252-254; Pg. 9 Line 267-268, Line 277-279, 285-288.
20. Conclusion: Please rewrite this part
Author response: Thank you for your feedback. We have edited the conclusion section as requested by the reviewer by adding and removing text as necessary. Pg. 10, Line 486-487 and 491.
21. Tables and Figures: Please have a look in the Results.
Author response: Thank you for your feedback. We have edited the figures section as requested by the reviewer. Pg. 6, Line 226, Pg. 7 and Pg. 11-12
22. References: Please harmonize the (accessed on Nov 21) and (accessed on 21 September 2017) all over the references. Be consistent in using abbreviations for journal’s names.
Author response: Thank you for your feedback. We have harmonized the reference formatting as requested by the reviewer ad in-line with the journal guidelines. Pg. 13-14.

This manuscript is a resubmission of an earlier submission. The following is a list of the peer review reports and author responses from that submission.

Round 1

Reviewer 1 Report

Thank you for a well written paper. Given the sampling frame ie: participants at a forum on lifelong vaccination, the results are hardly surprising. The commentary on the reasons why practitioners don't deliver on their apparent enthusiasm for the importance of lifelong vaccination is potentially valuable and could be further developed. Without this, the paper provides little in the way of direction about how LCI could be more comprehensively implemented.

It would be great to see this further developed because this is such an important issue given the worrying re-emergence of many previously eradicated infectious diseases

Reviewer 2 Report

This study presents the results of investigations on healthcare professionals’ vaccination throughout an individual’s lifespan or “life-course immunization (LCI)”. HCPs’ perceived importance of LCI implementation and barriers to implement LCI were demonstrated. In my perspectives, this study was more like a general survey, lacking of theoretical supports and having less contribution to both theoretical development and practical recommendation. To deal with a problem related to LCI implementation, it is better to deeply investigate underlying factors hindering the success of LCI implementation. Those are such as the government’s budget allocation, the general public’s awareness, national health policy, or lack of vaccine knowledge. This study just conducted a general survey by exploring HCPs’ opinions. Most importantly, those HCPs had different socio-economic characteristics and came from several regions, but these differences were not taken into the analysis at all. Consequently, the result cannot be generalizable and hard to convince policy makers for health policy development. Therefore, I cannot recommend publication of this manuscript. There are several methodological concerns and relevant issues as follows;

Data collection was conducted based on the surveys with participants of the 35th Annual Meeting of the European Society of Paediatric Infectious Diseases. Those participants came from several regions and had different socio-economic characteristics such as gender, working experiences, education level, age, etc. These factors could have a significant effect on HCPs’ opinions. Understanding this relationship could tell reliability of the results, and can imply how the results can be used for health policy development.

How were questions used for the surveys developed? The result of reliability test for the survey questionnaire was not shown, so the quality of survey instrument cannot be academically confirmed.

How many people participated in the electronic questionnaire survey and how many people were excluded?

The results came out quite general. For instance, a majority of participants “strongly agreed” or “agreed” on the need to position LCI. Most participants indicated that they were ready to discuss LCI with their patients and extend the recommendation for LCI to the patient’s family (59.4–75.7%). 77.4–88.6% indicated that LCI was a priority. No more details were explained.

I think this type of research should cooperate with results of in-depth interviews, so that, deep and clear explanations on explored situations can be shown.

In the result section, it is stated that “Interestingly, infectious disease and public health specialists also viewed LCI as a priority (77−88%)”. Please provide discussion.

It is also stated that participants indicated they encounter vaccine hesitancy in their practice but were not seen as the main barrier in the implementation of LCI. Please clearly explain this issue.

In my opinion, vaccine hesitancy can be one of important barriers. I am also curious whether population in more developed countries may have less vaccine hesitancy than those in developing countries. It would be interesting if this issue is discussed.

Overall, I think that the contribution of this research is so limited, and is not sufficient to confirm publication in this journal. My suggestion is that more investigations should be carried out. In the paper, it is also indicated that this survey has several limitations, and therefore the findings may not be generalizable.

Reviewer 3 Report

The authors should disclose whether there were presentations at the satellite symposium on the subject prior to eliciting survey responses, and if so, should share the full content of those presentations. The authors should mention, in the limitations, the lack of any discussion of risk-benefit and cost-benefit balance.

The authors should address the fact that a “lifetime immunization” approach is not a defined entity, and whether it is net favorable may depend on the specifics of the approach. For example, which immunizations are being advocated, to be given when, how often, and to whom?